# Moving beyond the age-depth model paradigm in deep sea palaeoclimate archives: dual radiocarbon and stable isotope analysis on single foraminifera.

Bryan C. Lougheed[1,2*], Brett Metcalfe[2,3*], Ulysses S. Ninnemann[4] and Lukas Wacker[5]

[1]Department of Earth Sciences, Uppsala University, Villavägen 16, 75236 Uppsala, Sweden.
[2]Laboratoire des Sciences du Climat et de l'Environnement, LSCE/IPSL, CEA-CNRS-UVSQ, Université Paris-Saclay, France.
[3]Department of Earth Sciences, Faculty of Sciences, Vrije Universiteit Amsterdam, De Boelelaan 1085, 1081HV Amsterdam, the Netherlands.
[4]Department of Earth Science, University of Bergen, Allégaten 41, 5007 Bergen, Norway.
[5]Laboratory for Ion Beam Physics, ETH Zürich, Otto-Stern-Weg 5, 8093 Zürich, Switzerland.

* contributed equally to this study.

*Correspondence to*: Bryan C. Lougheed (bryan.lougheed@lsce.ipsl.fr)

**Abstract.** Late-glacial palaeoclimate reconstructions from deep-sea sediment archives provide valuable insight into past rapid changes in ocean chemistry. Unfortunately, only a small proportion of ocean floor with suitably high sediment accumulation rate (SAR) is suitable for such reconstructions using the longstanding age-depth model approach. We employ ultra-small radiocarbon ($^{14}$C) dating on single microscopic foraminifera to demonstrate that the longstanding age-depth model method conceals large age uncertainty caused by post-depositional sediment mixing, meaning that existing studies may underestimate total geochronological error. We find that the age-depth distribution of our $^{14}$C-dated single foraminifera is in good agreement with existing bioturbation models when one takes the possibility of *Zoophycos* burrowing into account. To overcome the problems associated with the age-depth paradigm, we use the first ever dual $^{14}$C and stable isotope ($\delta^{18}$O and $\delta^{13}$C) analysis on single microscopic foraminifera to produce a palaeoclimate time series independent of the age-depth paradigm. This new state-of-the-art essentially decouples single foraminifera from the age-depth paradigm to provide multiple floating, temporal snapshots of ocean chemistry, thus allowing for successful extraction of temporally accurate palaeoclimate data from low SAR deep sea archives. This new method can address large geographical gaps in late-glacial benthic palaeoceanographic reconstructions by opening up vast areas of previously disregarded, low SAR deep-sea archives to research, which will lead to improved understanding of the global interaction between oceans and climate.

## 1 Introduction

The past seven decades in palaeoceanography research have produced a wealth of valuable palaeoclimate data from the calcareous, foraminiferal ooze contained in deep sea sediment archives, greatly improving our understanding of past ocean chemistry and palaeoclimate (Bond et al., 1993; Emiliani, 1955; Epstein et al., 1951; Shackleton, 1967; Urey, 1947). The

longstanding geochronological method that has been applied to these sediment archives since the inception of palaeoceanography as a field of study, the *age-depth model* method, relies on the geological law of superposition. This law states that sediment age increases progressively with sediment core depth. The age-depth model method, as applied to deep-sea sediment cores, involves first slicing deep sea sediment cores into discrete core depth intervals of 1 cm thickness or greater. Sufficient numbers of suitable foraminifera tests are subsequently picked from select intervals and analysed using mass spectrometry, whereby ages are inferred through [14]C dating and/or orbital tuning (Martinson et al., 1987; Pisias et al., 1984) of $\delta^{18}O$ values. Finally, statistical methods (Blaauw and Christen, 2011; Bronk Ramsey, 2008) are used to interpolate ages for all discrete depth intervals of the sediment core.

Post-depositional sediment mixing (PDSM) of deep sea archives (due to, e.g., bioturbation) can cause the relationship between sediment core age and sediment core depth to become more complex, thereby limiting the precise application of the law of superposition. This complexity can be masked from researchers who use the age-depth model method because successful stable isotope ratio mass spectrometry (IRMS) or [14]C accelerator mass spectrometry (AMS) analysis has traditionally required the analysis of multi-specimen samples containing tens (Metcalfe et al., 2015; Waelbroeck et al., 2005) and hundreds (Hughen et al., 2006) of single foraminifera tests, respectively. IRMS and AMS analyses only report a mean sample value and a machine measurement error, meaning that no information is provided about the true heterogeneity of the sample, which is chiefly a function of sediment accumulation rate (SAR) and PDSM. As such, the true age uncertainty within a sediment core can remain hidden from the researcher. Furthermore, AMS analysis reports a sample's mean [14]C *activity*, which is not the same as a sample's mean [14]C *age,* which creates further complications for highly heterogeneous samples (as alluded to by reviewer Ascough [2017]).

Considering the aforementioned complications associated with the age-depth model method, high resolution sampling of core depth does not necessarily translate into high resolution sampling of time. Researchers are aware that the concealment of intra-sample age heterogeneity can pose problems for the age-depth model method (Bard, 2001; Keigwin and Gagnon, 2015; Löwemark and Grootes, 2004; Löwemark and Werner, 2001; Pisias, 1983; Ruddiman and Glover, 1972), which can potentially lead to incorrect interpretation of temporal climate offsets, or apparent [14]C age offsets between different species and/or sizes of foraminifera that are, in fact, an artefact of PDSM (Berger, 1977; Löwemark et al., 2008; Löwemark and Grootes, 2004; Peng and Broecker, 1984). With these problems in mind, researchers seeking to reconstruct rapid (i.e. sub-millennial and centennial) climate processes have generally concentrated on sediment archives with a SAR greater than 10 cm/ka (Hodell et al., 2015; Shackleton et al., 2000; Vautravers and Shackleton, 2006), with the assumption that high SAR minimises the effects of PDSM upon age-depth models. However, the inability to directly quantify intra-sample age heterogeneity means that this assumption has yet to be rigorously tested. Furthermore, the vast majority of the ocean floor exhibits a SAR of less than 10 cm/ka (Fig. 1), meaning that many potentially useful study sites above the calcite compensation depth are essentially rendered unusable by the longstanding age-depth approach. Concentrating only on the select parts of the ocean floor with high SAR introduces a geographical bias into our understanding of global ocean processes.

In this study, we utilise the latest developments in ultra-small (<100 µg CaCO$_3$) sample $^{14}$C dating (Synal et al., 2007; Wacker et al., 2013a, 2013b, 2013c) to reduce sample size to a single benthic foraminifer specimen (*Cibicidoides wuellerstorfi*), thereby allowing us to directly quantify intra-sample age heterogeneity and analyse PDSM in the case of the low SAR (~1-2 cm/ka) sediment core T86-10P from the Azores region of the North Atlantic (Fig. 1). We discuss the consequences of our results for existing studies and provide suggestions for adding realistic geochronological errors to existing deep sea palaeoclimate records that have applied the longstanding age-depth model method using multi-specimen samples. Furthermore, we analyse both $^{14}$C and stable isotopes on select single foraminifer specimens of sufficient mass, demonstrating the feasibility to construct palaeoclimate time series that are completely independent of the age-depth paradigm and its associated problems.

## 2 Method

### 2.1 Sediment core selection and subsampling

Sediment core T86-10P (Fig. 1) was retrieved from the North Atlantic (37° 8.13' N, 29° 59.15' W, 2610 mbsl) by the vessel R/V *Tyro* as part of the APNAP-I project. We chose core T86-10P for this study because preliminary oxygen isotope measurements on planktonic foraminifera indicated a poor multi-specimen glacial-interglacial δ$^{18}$O stratigraphy, typical of a low SAR sediment core (Metcalfe, 2013). Also, the nearby presence of a very high SAR record at a similar water depth (Repschläger et al., 2015) provides an ideal local 'reference' stratigraphy for direct comparison. In other words, core T86-10P is an ideal sediment core with which to test the ability of dual $^{14}$C and stable isotope analysis on single foraminifera to successfully extract temporally accurate palaeoclimate data from a very low SAR archive.

single specimen benthic foraminifera tests (*Cibicidoides wuellerstorf,* the same species used in Repschläger et al. [2015]) ranging between 25 and 500 µg in mass were retrieved from the wet sieved and deionised water washed >250 µm fraction of 1 cm slices of the sediment core. 47 discrete 1 cm intervals were picked for foraminiera, with an average of 3.9 specimens per discrete interval being picked (min 1 specimen, max 10 specimens). Care was taken to pick whole tests that exhibited neither physical damage nor visible dissolution. Of these 185 tests, 100 were measured for stable isotopes only (δ$^{18}$O and δ$^{13}$C) and 49 were measured for $^{14}$C only, while 36 tests were successfully analysed for both $^{14}$C and stable isotopes. These 36 tests were cut using a scalpel (future work may use more efficient gaseous splitting) according to an approximate 80/20 ratio, with the larger fraction being reserved for $^{14}$C AMS analysis and the smaller fraction being reserved for stable isotope IRMS analysis. All data are available in spreadsheet format in Table S1.

### 2.2 $^{14}$C analysis

AMS analysis was carried out at the Laboratory for Ion Beam Physics at ETH Zürich using a permanent magnet equipped *Mini Carbon Dating System* (MICADAS) AMS (Synal et al., 2007) with Helium stripping system, coupled to an *Ionplus* carbonate handling system (Wacker et al., 2013a). The MICADAS setup allows for direct $^{14}$C measurement, using a gas ion

source (Wacker et al., 2013b), of gaseous $CO_2$ liberated from $CaCO_3$ samples by acidification with phosphoric acid – i.e. no graphitisation step is necessary. The exclusion of the graphitisation step allows for the required sample mass to be reduced down to 100 μg of $CaCO_3$ (12 μg C) and less (Lougheed et al., 2012; Wacker et al., 2013c), enabling sample size to be reduced to one specimen in the case of a suitable foraminifera species and specimen. Procedural single specimen foraminifera blank samples (assumed Eemian age) from core T86-10P indicated a mean blank value of 0.0115 $F^{14}C$ (n=10). Procedural IAEA-C1 standard blank material of similar mass as the single foraminifera tests yielded a mean blank value of 0.0100 $F^{14}C$ (n=10). No relationship was found between blank mass and blank value. Representative blanks (i.e. those not affected by flux jumps, etc.) were pooled and a specific blank correction of 0.0095 ±0.002 $F^{14}C$ was applied to most of the samples using the BATS software (Wacker et al., 2010). In the case of a small group of remaining samples, which came from a second run (those with lab code prefix ETH-74), a lower blank correction ($F^{14}C = 0.0033 ± 0.001$) was applied. Reported $^{14}C$ ages (Table S1) are rounded following standard conventions (Stuiver and Polach, 1977). Research is ongoing regarding achieving a superior blank value for such small samples, which would allow us to improve precision and further push back the age limit of single foraminifera $^{14}C$ analysis. The blank values we have achieved in this study, however, are sufficient for quantifying PDSM and reconstructing deglacial processes (i.e. the past 20 ka) in the low SAR core T86-10P. Our $^{14}C$ measurement precision is one order of magnitude smaller than the reconstructed PDSM, while the general geochronological uncertainties associated with $^{14}C$ age calibration of marine samples (i.e. reservoir age and calibration curve uncertainties) are generally greater than our measurement uncertainty. It could be argued that high precision $^{14}C$ analysis within the marine realm is not strictly necessary. Moreover, especially when studying highly heterogeneous material, sacrificing precision to reduce sample size to one specimen translates to more accurate results than when one carries out high precision measurements on multi-specimen samples – in the latter case one loses all information regarding the true age precision, i.e. the intra-sample age heterogeneity. We carried out experiments to investigate the possible presence of secondary carbonate phases in our samples; select single specimens (n=5) of sufficient size were $^{14}C$ analysed for both an initial phosphoric acid leach fraction and the residual foraminifer fraction. Of these five specimens, three specimens yielded leach and residual $^{14}C$ ages that were not significantly different within 1σ and a fourth specimen was not significantly different within 2σ (the fifth specimen was not significantly different within 2.1σ).

Benthic *C. wuellerstorfi* $^{14}C$ ages were calibrated using *MatCal 2.2* (Lougheed and Obrochta, 2016), employing the *Marine13* (Reimer et al., 2013) $^{14}C$ calibration curve and an appropriate marine reservoir age (ΔR=35±290 $^{14}C$ yr), the latter of which was calculated as follows: Previous studies in this region using planktonic foraminifera have employed the standard marine calibration curve (i.e. ΔR=0±0), but the possibility of spatiotemporally dynamic ΔR for the Azores region has been alluded to previously (Schwab et al., 2012; Waelbroeck et al., 2001). We are aware of the potential uncertainties associated with ΔR, so we employ a planktonic ΔR with a large uncertainty: 0±200 $^{14}C$ yr. Seeing as we carried out our investigation upon benthic foraminifera, we must additionally take into account the possibility that benthic ΔR may be different from that of the ocean's surface mixed layer for which *Marine13* was developed. Using available data from a nearby sediment core from the Azores region (MD08-3180) (Sarnthein et al., 2015), we analyse $^{14}C$ determinations from a late-glacial sequence of

co-occurring benthic and planktonic foraminifera (S1 and Fig. S1). We find that the long-term (7 ka) average [14]C age difference between the planktonic and benthic foraminifera is 35±210 [14]C yr, suggesting that there is only a small absolute difference between benthic and planktonic [14]C ages in this region, but with considerable variation. We arrive at our final benthic ΔR by correcting our planktonic ΔR (0±200 [14]C yr) for the benthic offset (35±210 [14]C yr) with error propagation, resulting in a final ΔR of 35±290 [14]C yr.

## 2.3 Stable isotope analysis

IRMS analysis on the smaller foraminifera test fractions from the cut tests (as well as some whole tests) was carried out at the stable isotope laboratory of the Department of Earth Science, University of Bergen, using a *Kiel IV* carbonate device coupled to a *Thermo MAT-253* dual inlet IRMS. The use of a dual-inlet IRMS, as opposed to a continuous flow IRMS, leads to a reduced size difference between sample and standard gas, combined with a continuous switching between standard and sample gas, which enables a higher analytical precision for small samples. Procedural standard samples (Carrara marble powder) of representative mass indicated an external precision (1σ) better than 0.10 ‰ and 0.05 ‰ for $\delta^{18}O$ and $\delta^{13}C$, respectively. Additional whole tests were analysed using a *GasBench II* preparation device coupled to a continuous flow *Thermo Delta Plus* mass spectrometer at the Department of Earth Sciences, Vrije Universiteit Amsterdam. For these measurements, the external precision (1σ) of international standards was better than 0.12 ‰ for both $\delta^{18}O$ and $\delta^{13}C$ (Feldmeijer et al., 2015; Metcalfe et al., 2015). All IRMS measurements are reported in per mil (‰) against the Vienna Peedee Belemnite (V-PDB) scale.

## 3 Results and Discussion

### 3.1 Age uncertainties concealed by the longstanding geochronological method

[14]C analysis carried out on 85 single specimen foraminifera tests for core T86-10P (Fig. 2A, Table S1) indicate the presence of significant PDSM, with the average standard deviation for all discrete 1 cm core depth intervals containing three or more [14]C dates being 4667 [14]C yr. We show that such significant PDSM can be concealed by longstanding geochronological methods. Specifically, we imitate the age-depth model approach involving multi-specimen samples by averaging all uncalibrated single specimen [14]C data from discrete depths with three or more measured single specimens into pseudo multi-specimen [14]C dates with an uncertainty of 60 [14]C yr (a typical AMS machine error for larger samples). We subsequently calibrate these pseudo multi-specimen dates and produce a Bayesian age-depth model for core T86-10P (Fig. 2B) using the *Bacon* software (Blaauw and Christen, 2011). This age model displays an apparent average SAR of 2.2 ±0.9 cm/ka, with higher apparent SAR in the uppermost 10 cm of the core than the lower parts of the core, which is consistent with typically observed sediment mixed layer depths (e.g. Trauth et al., 1997). Considering the large intra-sample heterogeneity present in core T86-10P, our pseudo multi-specimen Bayesian age-depth model exhibits unrealistically well-constrained confidence intervals, thus concealing the true age-depth variation present within core T86-10P. Additionally, the Bayesian age-depth

modelling routine excludes a number of pseudo multi-specimen dates as outliers. We propose that downcore multi-specimen [14]C dates in non-sequential temporal order may serve as useful indicators for the presence of significant PDSM throughout an entire sediment sequence, rather than being regarded as outliers related to isolated events, as is often done in the literature. While core T86-10P may not be representative of all sediment cores, the fact that very large intra-sample age heterogeneity can be concealed by longstanding geochronological methods has significant implications for existing studies relying on the age-depth model method. We note that our pseudo multi-specimen [14]C dates were assembled using data from an average of four foraminifera tests each. Typically, multi-specimen samples containing many tens to hundreds of foraminifera tests are used for [14]C dating. Were such large sample sizes to be applied to T86-10P, it is possible that no age-depth outliers would be produced and that all information about intra-sample heterogeneity would be lost, thus concealing the full temporal uncertainty from the researcher. Such an affect was seen in one of the earliest studies using ultra-small [14]C dating of foraminifera samples (Lougheed et al., 2012), whereby downcore age-depth reversals were found for a sequence of multi-specimen samples with <500 µg mass, whereas a sequence of multi-specimen samples with a greater mass did not exhibit such a behaviour. We carried out simulations to investigate the influence of [14]C sample size upon the concealment of age-depth outliers by using multiple simulated synthetic sediment core scenarios whereby exaggerated single foraminifera PDSM is generated using Gaussian noise (S2 and Fig. S2). These preliminary simulations suggest that when sample size is five to ten specimens or more, no age-depth outliers are present in a simulated sediment core with intense PDSM.

## 3.2 Core T86-10P in the context of existing bioturbation understanding

Seeing as it has not been previously possible to [14]C date single foraminifera, past research of PDSM has focussed on quantifying bioturbation processes by constructing theoretical models. One of the earliest models for bioturbation in deep sea sediment cores by Berger and Heath (1968) assumes that the uppermost layer of sediment core is uniformly to a certain mixed layer depth (typically 10 cm) throughout the sedimentation history of the core. This general model of bioturbation has formed the basis for subsequent modelling investigations into vertical PDSM of sediment particles of different age (e.g., Berger and Johnson, 1978; Berger and Killingley, 1982; Guinasso and Schink, 1975; Peng et al., 1979). A general feature of these traditional bioturbation models is that the continuous application of a uniformly mixed upper layer of sediment throughout the entire sedimentation history of given core archive will result in an exponential probability density function (PDF) for age at any given core depth. Such an exponential PDF will exhibit a maximum probability for younger ages, with a long tail towards older ages.

To determine if the single foraminifera [14]C data we retrieved from core T86-10P can be approximated using the aforementioned bioturbation models, we have carried out a single foraminifera sedimentation simulation using a uniform mixed layer depth of 10 cm and constant SAR of 1.4 cm/ka. The SAR applied is on the low end of the SAR estimated by the *Bacon* age-depth model (2.2 ±0.9 cm/ka), because we have considered that much of the *Bacon* age model also includes the interval of the sediment core within the active mixed layer depth. We consider, therefore, that 1.4 cm/ka represents a good estimation the core SAR outside the mixed layer depth (i.e. the SAR corresponding to the historical layers). Our simulation

is carried out in sedimentation intervals of 0.1 cm, with 400 new foraminifera being added per interval and assigned an age according to the SAR. At each sedimentation interval, the upper 10 cm of the sediment is uniformly mixed using random noise. The results of the simulation are presented in Figs. 3A and 3B, and superimposed upon them are the calibrated ages for all single foraminifera that we have [14]C dated. The simulation would seem to suggest that the population of single foraminifera we have [14]C dated in our study cannot be approximated using a sedimentation simulation that uses only a uniformly mixed layer depth, i.e. the method used by traditional bioturbation models. While we apply a constant SAR, using a dynamic SAR in our simulation would not resolve this disagreement, because it would simply shift the median age towards certain [14]C-dated single foraminifera, but away from others (Fig. 3B). It is apparent that the age-depth distribution of our single foraminifera is not compatible with the exponential age PDF predicted by traditional bioturbation models. Most notably, such bioturbation models cannot explain the downward movement of young foraminifera to great depth in T86-10P, as in the case of the specimen at 41-42 cm depth (7.9 cal ka) or the one at 21-22 cm depth (1.2 cal ka) (Fig. 3B).

To further analyse the age-depth distribution of the [14]C dated single foraminifera in T86-10P, we construct an ideal superposition ranking of the single foraminifera (i.e. in the case of no PDSM) from core T86-10P by ranking them by median calibrated [14]C age. We can also rank the foraminifera by sediment core depth, i.e. by the 1 cm depth interval they were actually retrieved from. 'Depth ranking' is not the same as 'depth', but nonetheless provides an interesting way to visualise the movement of single foraminifera. By comparing the age ranking and depth ranking, we can visualise the post-depositional upcore and downcore movement of the single foraminifer tests (Fig. 4A). Analysis of the post-depositional ranking change for the foraminifera indicates that while the ranking change appears normally distributed, it just fails a statistical test for normal distribution (Fig. 4B). It would appear, therefore, that the age distribution of the single foraminifera that we have [14]C dated cannot be fully approximated using a normal distribution, nor by the exponential distribution suggested by traditional bioturbation models.

It may be possible that secondary bioturbation processes are contributing to the observed PDSM in T86-10P. The burrows of an unidentified organism, referred to as ichnofacies *Zoophycos* burrows, has been shown to penetrate much farther down into the sediment than the uniform mixed layer depth (Löwemark and Werner, 2001; Wetzel and Werner, 1980). These burrows are often invisible to the naked eye, with X-ray radiographs being necessary for identification. Such secondary bioturbation effects are not considered by traditional bioturbation models, a fact that researchers have previously noted (Löwemark and Grootes, 2004; Löwemark and Werner, 2001). The practical effect of *Zoophycos* secondary bioturbation upon a given discrete depth interval would be to introduce a population of significantly younger sediment from above (Löwemark and Schäfer, 2003), thus altering the age distribution of that discrete depth interval. The potential presence of *Zoophycos* may, therefore, offer an explanation for the apparent disagreement between the single foraminifera [14]C age-depth relationship for core T86-10P and that predicted by our sedimentation simulation when only the traditional model of bioturbation is applied (Fig. 3B). To further investigate this possibility, we carry out a new single foraminifera sedimentation simulation in Figs. 3C and 3D. This simulation is forced using the same SAR (1.4 cm/ka) and mixing layer depth (10 cm) parameters as the simulation previously described in Figs. 3A and 3B, but with the addition of a post-simulation bioturbation by *Zoophycos*.

Specifically, we take 10% of the total single foraminifera population from the entire core and add an additional randomly selected depth between 0 and 50 cm to their depth values. This process essentially serves to simulate 10% of the core sediment being bioturbated downwards by *Zoophycos*. The addition of the *Zoophycos* procedure to our simulation produces a simulated foraminifera age-depth relationship that can reconcile the presence of the young [14]C-dated foraminifera that we find at depth in core T86-10P (Fig. 3C and 3D).

### 3.3 Consequences for existing studies using longstanding geochronological methods

The concealment of PDSM by longstanding geochronological methods presents significant consequences for existing studies that use stratigraphic and geochronological data sourced from deep sea sediment archives to reconstruct rapid changes in palaeoclimate. Many recent such studies (e.g., Barker et al., 2015; Caley et al., 2012; Simon et al., 2016) use tuning to the LR04 (Lisiecki and Raymo, 2005) benthic stack to produce an age-depth chronology. We have rerun our single foraminifera sedimentation simulation using the average SAR of LR04 (3.9 cm/ka) and found that one could expect a relative 68.2% age range of -1330 to 2950 yr for discrete 1 cm depth intervals. When tuning to LR04, one must also consider any uncertainties related to the tuning process itself, which may be on the order of multiple millennia (Blaauw, 2012; Martinson et al., 1987; Pisias et al., 1984).

For some continental margin sites, such as those from the Iberian Margin, SAR is very high (20-30 cm/ka) and such study sites have been used for centennial resolution age-depth climate reconstructions (i.e. ±50 years precision), with the assumption that high SAR essentially minimises the effect of bioturbation upon age-depth reconstructions (Hodell et al., 2015; Shackleton et al., 2000; Vautravers and Shackleton, 2006). On the other hand, Bard (2001) suggests that such high SAR can at best be used only for millennial resolution (i.e. not centennial). We have rerun our single foraminifera sedimentation simulation using a high SAR (20 cm/ka) typical of Iberian Margin sites (Figs. 3E and 3F). These simulations suggest a relative 68.2% age range of -260 to 570 yr for a discrete 1 cm depth interval.

Specifically in the case of [14]C dating of multi-specimen samples, it is of great important to consider the heterogeneity of the age distribution of a discrete sediment interval. Radiocarbon laboratory results are based on the mean [14]C activity of a sample, to which laboratories apply the Libby half-life in order to report a radiocarbon age in [14]C years (hereafter, "AMS age"). [14]C is a radioactive isotope and its activity relationship with time is exponential, with the consequence that a highly heterogeneous multi-specimen sample may produce an AMS age that is significantly offset from the actual mean [14]C age of the sample. For all three sedimentation scenarios in Figs. 3A, 3C and 3E, we have calculated the relative mean age, median age and AMS age for discrete depth intervals in all three simulation scenarios, allowing us to demonstrate the behaviour of the potential AMS age bias caused by heterogeneous age distributions. Comparing the simulations in Fig. 3A and 3C, it is apparent that the addition of only a relatively small amount of younger material with an exponentially higher [14]C activity can serve to significantly shift the AMS age towards a much younger value. Researchers carrying out both [14]C AMS and stable isotope ($\delta^{18}O$ and $\delta^{13}C$) analysis on the same sediment core should, therefore, be aware that the AMS age is potentially skewed by an activity bias, while stable isotopes are not. It is paramount, therefore, to consider the effect of discrete depth

interval age distribution when carrying out palaeoclimate reconstructions. It is furthermore possible that the discrete depth age distribution of a particular foraminifera species can change throughout the history of a sediment archive as a result of temporal changes in SAR, species abundance, mixed layer depth, *Zoophycos* intensity, etc.

It must be stressed that core T86-10P represents a single sediment archive location and may not be wholly representative for all locations. Moreover, our study is based on the analysis of *C. wuellerstorfi* within the >250 μm fraction, whereas foraminifera in smaller fractions may be differently affected by PDSM (Wheatcroft, 1992). An exact quantification of the intra-sample age heterogeneity at other locations is essentially unknown because it can be concealed by longstanding geochronological methods. Furthermore, the intra-sample age heterogeneity for less consolidated sediment within actively bioturbated younger sediment sequences may differ from the intra-sample age heterogeneity for older, more consolidated sediment. We propose, therefore, that the ultra-small sample [14]C methods we outline in this study can be used, in combination with modelling techniques, to help quantify intra-sample age heterogeneity for various sediment sequences at other study locations (including those in the LR04 benthic stack), thus allowing for the application of a suitable downcore geochronological uncertainty. Such an approach will ultimately lead to a better temporal integration of deep sea sediment archives within the global palaeoclimate record.

## 3.4 Bypassing the age-depth model paradigm?

We show that the limitations of the age-depth model paradigm essentially preclude the extraction of temporally useful deglacial benthic ventilation data from very low SAR archives such as core T86-10P. The traditional, discrete depth average (multi-specimen) downcore stable isotope stratigraphy for core T86-10P (Figs. 5A and 5B) shows many spurious, large excursions in $\delta^{18}$O and $\delta^{13}$C. The underlying cause for these large excursions is revealed by single specimen foraminifer $\delta^{18}$O and $\delta^{13}$C data (Figs. 5A and 5B), which show a large spread in values, an artefact of PDSM. This spread in values is significantly larger than the machine error associated with IRMS analysis (typically 0.1‰), meaning that it would be concealed by multi-specimen sample IRMS analysis.

Dual measurements of both [14]C and stable isotopes ($\delta^{18}$O and $\delta^{13}$C) on the same single specimen foraminifer in core T86-10P can decouple an individual foraminifer from the sediment archive to provide a floating, temporal snapshot of ocean chemistry that is independent of the geological law of superposition, thereby contributing palaeoclimate information wholly independent of the age-depth paradigm. Multiple such snapshots can facilitate a time history of ocean chemistry that is completely insensitive to PDSM induced issues involving multi-specimen samples within the age-depth paradigm, such as spurious age artefacts between foraminifera of different species, abundance changes, SAR changes, morphologies, dissolution/preservation conditions, particle size dependent mixing, etc. (Berger, 1977; Löwemark et al., 2008; Löwemark and Grootes, 2004; Peng and Broecker, 1984; Wheatcroft, 1992).

Due to the combined measurement size requirements of both AMS and IRMS, our dual [14]C and stable isotope measurements on single *C. wuellerstorfi* specimens were limited to those of sufficient mass (>100 μg $CaCO_3$), which are generally less abundant during the coldest stadial conditions, such as the last glacial maximum (LGM); a problem that also affects studies

using traditional, multi-specimen reconstructions (e.g., Shackleton et al., 2000). We were able to produce successful dual [14]C and stable isotope measurements for a sufficient number of foraminifera, revealing a *C. wuellerstorfi* benthic deglaciation signal for core T86-10P. This benthic deglaciation signal is in good agreement with existing *C. wuellerstorfi* data from a previous study using a nearby (140 km proximity) high SAR (~20 cm/ka) record (Repschläger et al., 2015) (Figs. 5C and 5D). Specifically, we find good temporal agreement with the absolute values for $\delta^{18}O$, indicating a valid benthic deglaciation signal. We also find good temporal agreement with a sharp peak in $\delta^{13}C$ values that has previously been interpreted as a local increase in Eastern North Atlantic Deep Water (ENADW) linked to the onset of the Holocene (Repschläger et al., 2015). Our results demonstrate that it is possible to use our dual [14]C and stable isotope method on single foraminifera to extract temporally accurate deglacial benthic palaeoceanographic data from a very low SAR site, with success comparable to a very high SAR site where traditional methods were used.

## 4 Conclusion

Analysis of [14]C on single foraminifera opens up new possibilities for quantifying the total geochronological error in existing studies due to PDSM. These errors may have been previously overlooked due to inherent limitations associated with the longstanding geochronological method based on multi-specimen species within the age-depth paradigm. Using the methods outlined in this study it is possible to quantify the age-depth geochronological uncertainty for existing late-glacial palaeoceanographic records, as well as to consider the possibility of AMS age biases associated with very heterogeneous multi-specimen samples. Full consideration of uncertainties will help to place existing palaeoceanographic records within an accurate geochronological framework. Subsequent improved evaluation of perceived regional leads and lags in palaeoceanographic processes will lead to an improved understanding of rapid climate change.

We also demonstrate that dual [14]C and stable isotope ($\delta^{18}O$ and $\delta^{13}C$) measurement on single foraminifera can produce temporally accurate benthic ocean chemistry data that is independent of the age-depth paradigm. This development opens up many new avenues in late-glacial palaeoceanographic research, specifically for the vast low SAR areas of the ocean (<10 cm/ka; Fig. 1) that are inaccessible for research using existing methods, thus filling in large spatial gaps in the global, late-glacial climate record. The resulting improvements in spatiotemporal reconstructions of global benthic ventilation conditions of the ocean across the glacial-interglacial transition will help to better understand the interaction between the atmosphere and ocean during periods of rapid climate change.

Retrieving the entire deglacial signal from low SAR sites using our proposed dual [14]C and stable isotope method may also prove to be cost effective, seeing as less elaborate sediment retrieval methods are necessary. Only the top 10-20 cm of the sediment archive are required and the preservation of sediment superposition is not of importance, so it may be possible to retrieve suitable sediment from the low SAR areas of the ocean simply by using grab samples, including those already present in institutional archives. Furthermore, the method we used to [14]C-analyse single foraminifera is efficient and cost-

effective, for two main reasons: (1) the elimination of the graphitisation process reduces labour and material costs; (2) the very small sample mass means that the required AMS machine analysis time is greatly reduced.

**Author contributions:** BCL and BM designed the study. BM picked and cut suitable foraminifera tests. LW and BCL carried out [14]C dating. BM, USN and BCL carried out stable isotope analysis. BCL analysed the data and wrote the manuscript with input from the co-authors.

**Acknowledgments:** Measurements and BCL were funded by Swedish Research Council (Vetenskapsrådet) grant 637-2014-499 awarded to BCL. BM acknowledges the Netherlands Organisation for Scientific Research (NWO) grant 822.01.0.19. Gerald M. Ganssen is thanked for providing access to core material. Claire Waelbroeck is thanked for assistance in finding planktonic-benthic [14]C ventilation data from the Azores region. We thank the valuable contributions of an anonymous reviewer and reviewer Philippa Ascough, as well as the valuable *Climate of the Past Discussions* contributions from Andrew M. Dolman, Sze Ling Ho, Thomas Laepple and Julia Gottschalk. Kevin Küssner is thanked for interesting discussions about *Zoophycos*. The town of Port Aransas, where most of this manuscript was written, is wished all the best in its recovery from Hurricane Harvey.

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

465

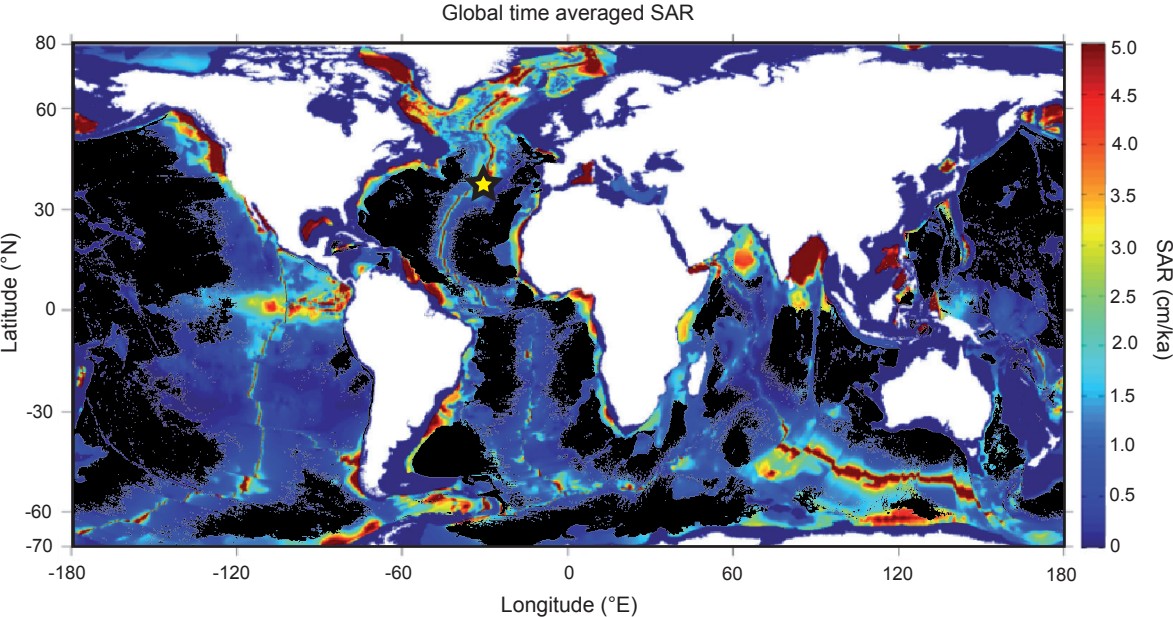

**Fig. 1.** Map of time averaged deep sea sediment accumulation rates (SAR) adapted from Olson et al. (2016). Note that continental margins are not included in the SAR estimate. Superimposed on this map in black is an approximate indication of seafloor areas under the calcite compensation depth, estimated using a global water depth of 4500 m derived from global bathymetry (General Bathymetric Chart of the Oceans, 2015). The location of core T86-10P (37° 8.13' N, 29° 59.15' W, 2610 mbsl) is indicated by a yellow star.

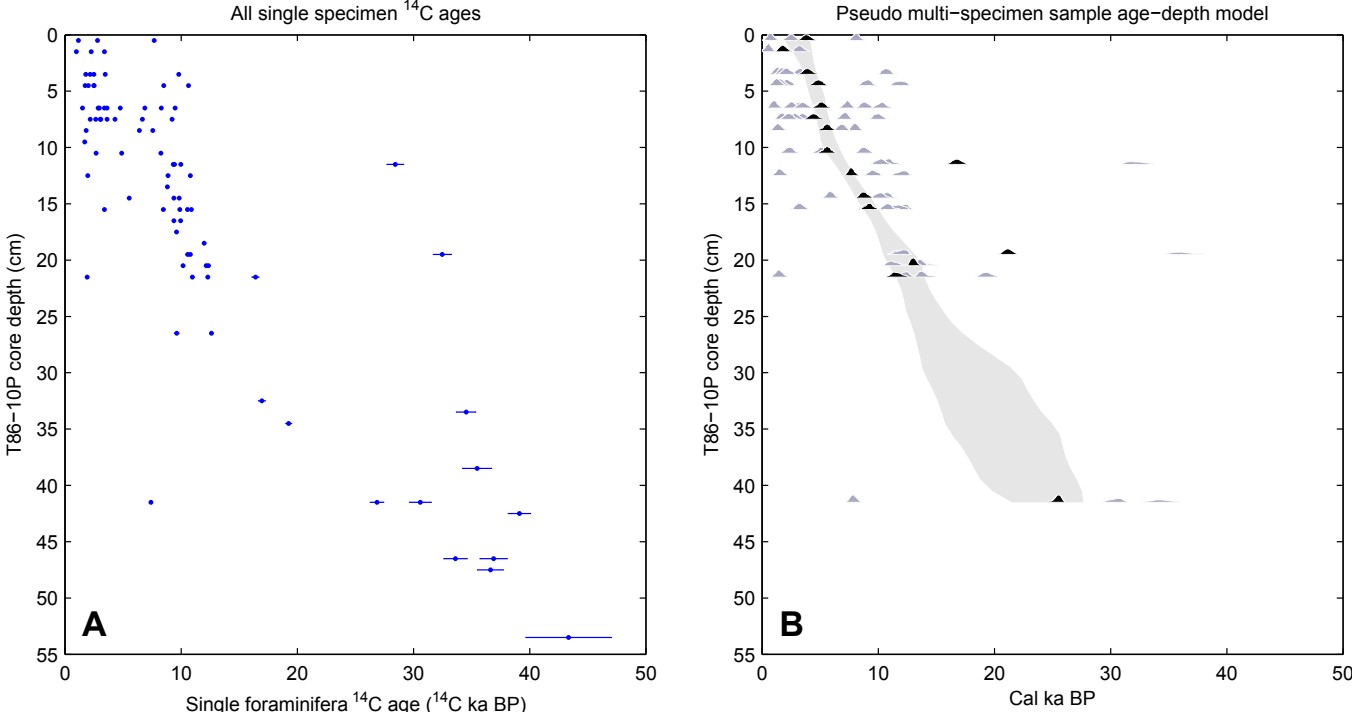

**Fig 2.** Results of single foraminifera *C. wuellerstorfi* [14]C dating in core T86-10P. **(A):** [14]C determinations (uncalibrated) on single *C. wuellerstorfi* foraminifera from core T86-10P, with 1σ measurement uncertainty shown. **(B):** A demonstration of the concealment of large intra-sample heterogeneity by the longstanding geochronological method. Calibrated single specimen benthic foraminifera [14]C ages (light blue probability density functions [PDFs]; see Method for [14]C calibration process) for all discrete 1cm core depths with three or more subsampled foraminifera. To demonstrate the potential concealment of PDSM by longstanding geochronological methods involving dating of multi-specimen foraminifera samples, we also show the 95% confidence interval (light grey) of a *Bacon* (Blaauw and Christen, 2011) age-deph model created using calibrated pseudo multi-specimen foraminifera [14]C ages (black PDFs – see text 3.1).

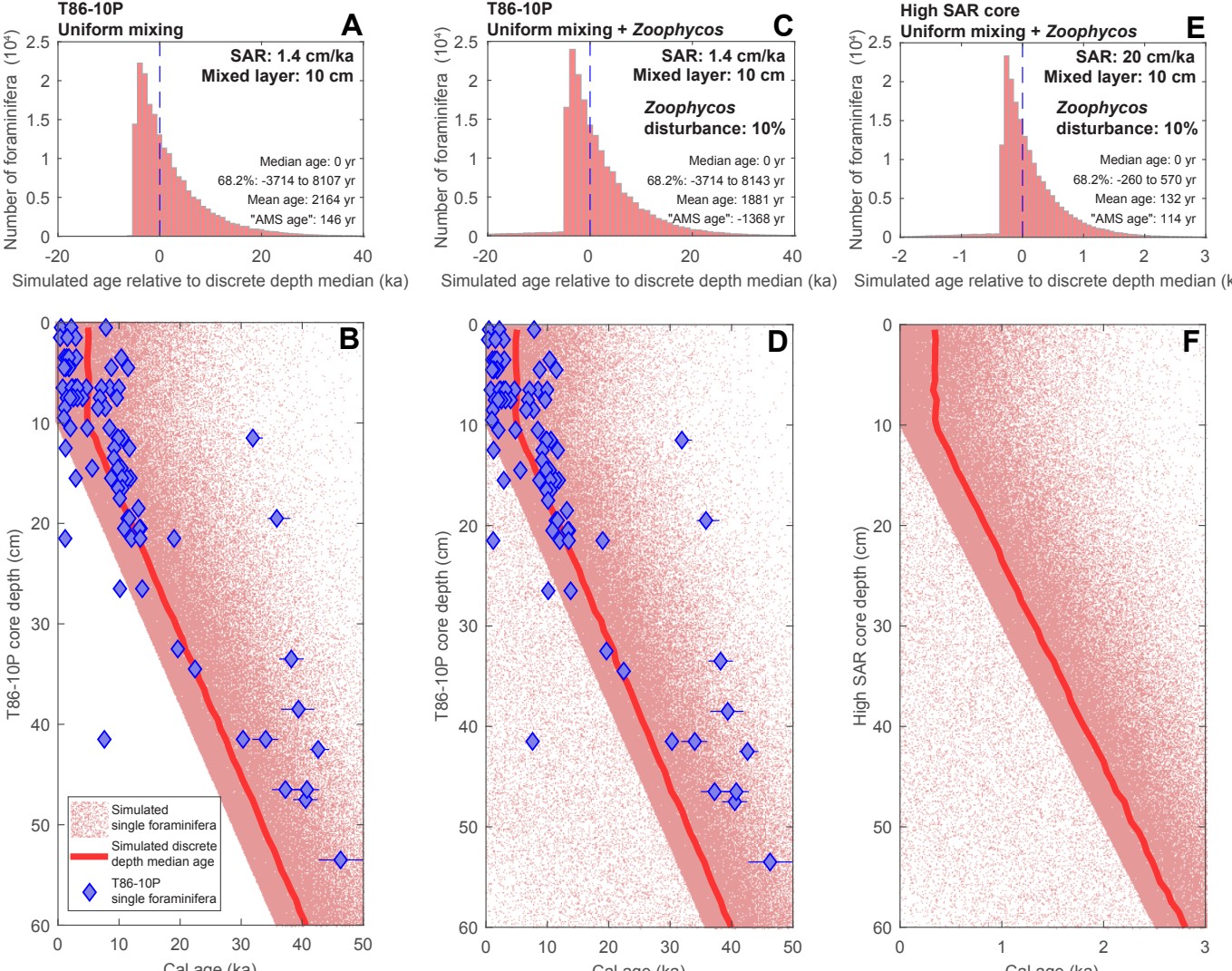

**Fig 3.** In order to investigate the age distribution for discrete depths of a sediment core, we carried out multiple single foraminifera sedimentation scenarios. Each of the simulations assumes a constant SAR, with foraminifera of the same size and species. Each scenario begins at 150 cm core depth and continues in 0.1 cm intervals to 0 cm depth. At each 0.1 cm interval, 400 new synthetic foraminifera are created and assigned an age according to the SAR. At every 0.1 cm interval, the uppermost 10 cm (the assumed mixed layer depth) of synthetic sediment core is uniformly mixed using random noise. While she simulations start at 150 cm, we only show data to 60 cm depth, in order to rule out any artefacts due to simulation spin-up. **(A), (B):** Single foraminifera simulation using an assigned SAR of 1.4 cm/ka and mixed layer depth of 10 cm. The scatter plot in panel B shows all simulated single foraminifera, along with the simulated discrete depth (1 cm depth slice) median age. Superimposed are the calibrated ages of the actual single foraminifera that we analysed in T86-10P. Histogram in panel A represents the age value of all simulated single foraminifera between 0 and 50 cm relative to their associated discrete 1 cm depth interval median age. The 68.2% percentile range, mean, median and AMS age is calculated for the histogram. AMS age is calculated by converting every single foraminifera age value to $F^{14}C$, from which a mean $F^{14}C$ value is calculated and subsequently converted back to $^{14}C$ yr. For these first-order AMS age estimations, we assume that $^{14}C$ years are equal to calendar years. **(C), (D):** As for panels A and B (SAR 1.4 cm/ka and mixed layer depth of 10%), but with added simulation of *Zoophycos* burrows. *Zoophycos* is added post-simulation by taking 10% of all foraminifera from all depths and increasing their depth value with a random value between 0 and 50 cm. **(E), (F):** High SAR single foraminifera sedimentation simulation (20 cm/ka and 10 cm mixed layer depth) with *Zoophycos* added in the same manner as for panels C and D.

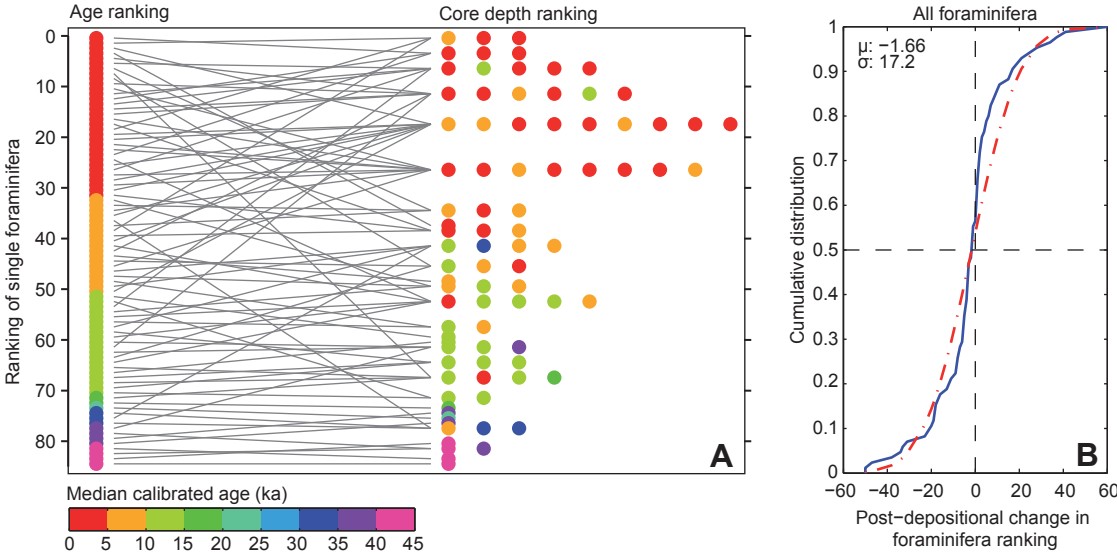

**Fig 4. (A):** Visualization of PDSM in core T86-10P. Single *C. wuellerstorfi* benthic foraminifera (represented by colored dots coded by median age) from core T86-10P are ranked by median calibrated age (left) and by core depth (right). Single foraminifera recovered from the same core depth interval are given the same core depth ranking. Grey lines visualize the reconstructed post-depositional change in ranking. **(B):** Cumulative distribution plot of ranking change (as ascertained from Fig. 3A) for single *C. wuellerstorfi* benthic foraminifera. The blue line represents an empirical cumulative distribution of the ranking change data. The broken red line represents a fitted normal cumulative distribution function (CDF) based on the mean and standard deviation of the ranking change data. Kolmogorov–Smirnov (K-S) testing is used to test a null hypothesis of the fitted normal distribution being similar to the empirical data. Hence, a P value *greater* than 0.05 (i.e. *not* less than or equal to) indicates normal distribution of the data at the α=0.05 significance level. K-S test result: P=0.047; not greater than 0.05.

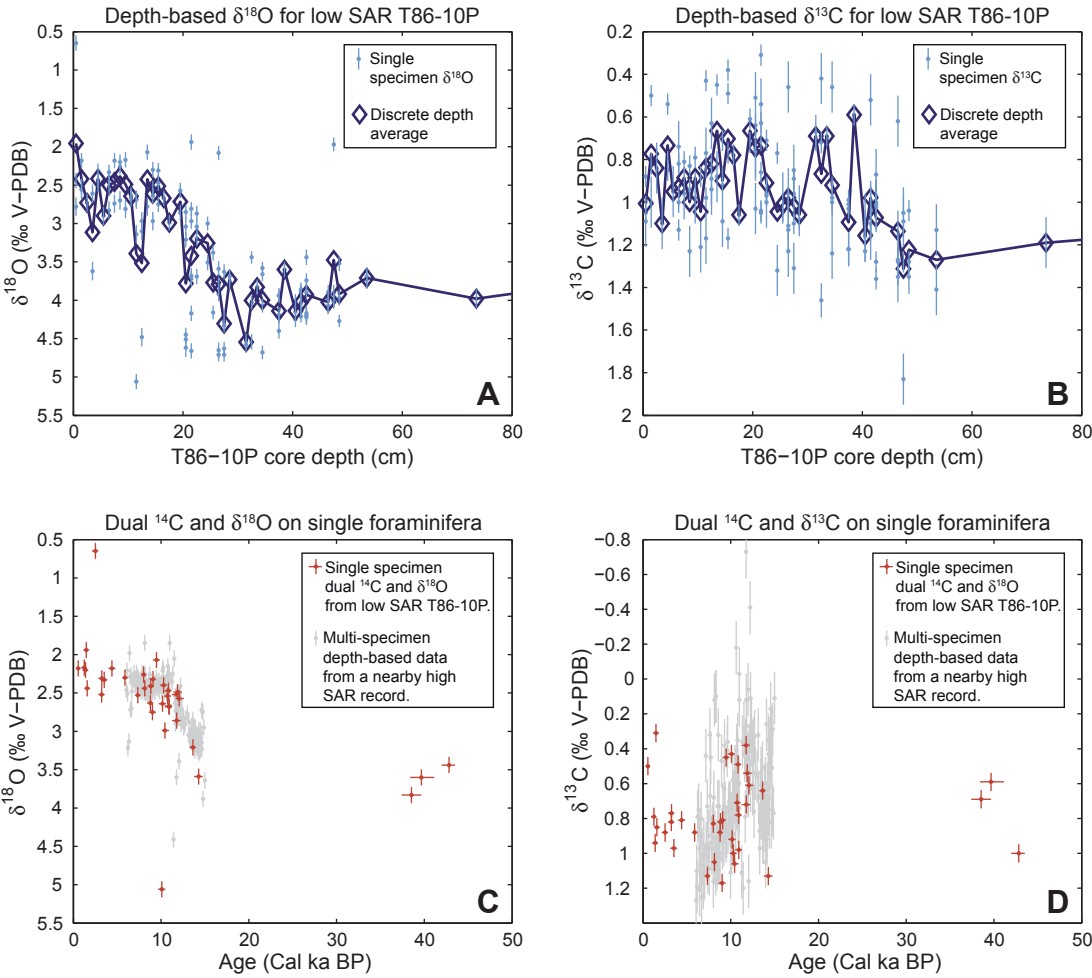

**Fig. 5.** Demonstration of successful use of dual $^{14}$C and stable isotope analysis on single foraminifera to retrieve useful paleoclimate information from the low SAR core T86-10P. **(A):** To visualise the inability to retrieve useful glacial-interglacial paleoclimate data from the low SAR core T86-10P using longstanding geochronological methods, we show single specimen *C. wuellerstorfi* $\delta^{18}$O data against core depth (with 1σ measurement error). Also shown is the average $\delta^{18}$O value of the single specimens for each discrete sediment core depth analysed (i.e. representative of longstanding methods using multi-specimen samples). **(B):** Same as for panel A, but with $\delta^{13}$C instead of $\delta^{18}$O. **(C):** Successful dual $^{14}$C and $\delta^{18}$O measurements on single foraminifera from core T86-10P. Vertical error bars denote 1σ error measurement. Horizontal error bars denote the 68.27% highest posterior density (HPD) interval of the calibrated $^{14}$C age (see Method for $^{14}$C calibration process). Also shown are previously published multi-specimen $\delta^{18}$O data from a nearby high SAR (20 cm/ka) record (Repschläger et al., 2015). Vertical error bars represent 1σ measurement error. **(D):** Same as for panel C, but with $\delta^{13}$C instead of $\delta^{18}$O.