# Peer review of "Moving beyond the age-depth model paradigm in deep sea palaeoclimate archives: dual radiocarbon and stable isotope analysis on single foraminifera."

_Climate of the Past, 2017_

## Author Comment (AC1) · 3 Oct 2017

Dear reviewers, readers and potential discussion participants, please note that there is an error in the caption of Figure 3 that could cause some confusion.

The following sentence:

"Cumulative distribution plots of ranking change (as ascertained from **Fig. 4**) for single C. wuellerstorfi benthic foraminifera."

should read as follows:

"Cumulative distribution plots of ranking change (as ascertained from **Fig. 3A**) for single C. wuellerstorfi benthic foraminifera."

Kind Regards, Bryan Lougheed

---

## Short Comment (SC1) · 27 Oct 2017

Dear Authors,

I have read your CPD contribution with great interest. Your study site is "ideal" to emphasize the influence of bioturbation on proxy data, and I agree that it is important to correct for these biases with adequate approaches (you present a novel and elegant one).

As there is very little specific information on sample sizes and data correction/postprocessing in the text and in the supplement (e.g., in Table S1), I wonder how contamination (that has increasing impact with smaller sample size) contributes to the 14C age differences of samples observed in your core (e.g., Brown and Southon, 1997; Hua et al., 2004). If corrections have been made without considering a size-dependent influence from contamination, different sample aliquots with a large size range (from ultra-small to normal size) can have significantly different 14C ages. This difference of course depends on the range of the sample sizes and the age of the sample, and will not compensate for the large bioturbation-driven 14C differences you observe. However, given the absence of specific information on samples sizes, I wonder whether it plays a role, in particular for some data points in Figure 2A. The blank seems to have been obtained on small samples ($\sim$50 $\mu$g C) so that data of samples with a size very different from the blank may be slightly over- or undercorrected on the basis of a size-independent, constant blank. Could you provide information on samples sizes and whether correction for contamination was applied or was not required to stress that 14C age differences in sediment core T86-10P are driven by bioturbation?

References

Brown, T.A., Southon, J.R., 1997. Corrections for contamination background in AMS 14C measurements. Nucl. Instruments Methods Phys. Res. Sect. B Beam Interact. with Mater. Atoms 123, 208–213. doi:10.1016/S0168-583X(96)00676-3

Hua, Q., Zoppi, U., Williams, A.A., Smith, A.M., 2004. Small-mass AMS radiocarbon analysis at ANTARES. Nucl. Instruments Methods Phys. Res. Sect. B Beam Interact. with Mater. Atoms 223-224, 284–292. doi:10.1016/j.nimb.2004.04.057

---

## Referee Comment (RC1) · Dr Ascough (Referee) · 29 Oct 2017

Review of cp-2017-119

The authors are to be commended on a careful piece of work, well within the remit of the journal. It is impressive analytically to obtain secure results on such small samples, and this study will be a useful addition to the literature.

In a composite date, if the preponderance of measurements were representative of the age of the sediment (albeit with a long tail), this would increase the accuracy of the

date, as the inaccurate ages would contribute proportionally less to the composite age. Can the authors comment on any statistical manipulation that could be used in this instance to improve accuracy? Also, how would one properly calculate (even semi-quantitatively) the uncertainties associated with composite measurements? Can the authors offer any suggestions for this? It would be good to see these points considered within the manuscript.

The addition of older material would have a proportionally smaller effect on the measured age than the addition of (much) younger material. Can the authors comment on how this would affect how the precision on measurements should be calculated (mass balance approach?).

One interesting point is that perhaps the changes in PDSM that could be identified with this approach could tell us something about sediment dynamics. I appreciate this is a little outside the scope of the paper, but could be mentioned as a 'silver lining'.

One reference that would be good to include is: Berger, W.H. and Johnson, R.F. 1978: On the thickness and the 14C age of the mixed layer in deep sea carbonates. Earth and Planetary Science Letters 41, 223–27. The findings in this support the authors results

From line 217 on- give more information about the palaeoclimate reconstructions and their significance- I appreciate that is not the thrust of this paper, but it would be useful to highlight the utility of the author's proposed approach.

How many samples would the authors advocate measuring in order to get a decent idea of the true amount of sample heterogeneity?

• State how many different layers the forams were selected from? How many forams per layer? • It might be good to expand a little on the evidence for PDSM in core T86-10P, just to give the reader an idea of what was observed • Were the forams pretreated in any way? i.e. washing/ agitation with distilled water, or removal for surface C by preliminary acid dissolution? • How were the measurements background corrected (i.e. explicitly state what blank was used) • Line 112. IRMS spelled incorrectly • Line 142. Effect • Line 39: analysis of multi-specimen • Line 38: Accelerator Mass Spectrometry •

---

## Short Comment (SC2) · 16 Nov 2017

Comment: Andrew M Dolman, Sze Ling Ho, Thomas Laepple.

We agree with the authors that the effects of post-depositional sediment mixing (PDSM), i.e. bioturbation, are important to consider and are hidden by conventional dating methods that measure a single 14C value from a sample containing many discrete specimens. As the number of distinct, e.g. forams, required for a 14C sample decreases – the question of how representative these few individuals will be of the layer

from which they were recovered becomes increasingly important. This uncertainty will not be measured unless multiple replicate samples are dated from the same sediment layer. By taking this to the extreme and dating individual forams, Lougheed et al are able to directly estimate the standard deviation in age of individuals recovered from the same 1 cm sediment layer atÂǎ4670 14C years. Widespread use of individual dating would enable a much more robust consideration of the effects of bioturbation on proxy records. In fact, by using a particularly large species of foram, they take this one stage further, and perform 14C dating on just half of each foram test, allowing d18O to be measured on the remaining half. This removes the need for depth-age modelling altogether and we agree that this is indeed a very exciting possibility that will open up regions of low sedimentation rate to proxy climate reconstruction with high temporal resolution.

Lougheed et al also simulate sediment cores for a range of sediment accumulation rates and a large number of PDSM intensities. The PDSM is modelled by applying Gaussian noise to the position of forams in the sediment cores with a range of standard deviations. Their figure 4 shows the results of the simulation. Using the observed sediment accumulation rate of 2.2 cm/ka, they estimate the standard deviation of movement required in order to obtain a SD of age of 4670 14C years as approximately 6 cm. We note that in the region of the observed sediment accumulation rate for this core, the contour line for a SD of $\sim$ 5000 14C years is more or less parallel to the PDSM axis and therefore there is little power to constrain the strength of PDSM. Also, if a constant sedimentation rate is assumed, then is the SD in depth not simply the SD in time (4670 years) scaled by the sedimentation rate, i.e. 4670 * (2.2 / 1000) = 10.274 cm?

We think this paper could be improved by instead considering a well established physical model of bioturbation in which there is a well-mixed surface layer of sediment of a fixed depth (Berger & Heath, 1968). Assuming a constant sedimentation rate, and a fixed mixing depth, the time integrated solution to this simple model predicts that the ages of material at a given depth follow an exponential distribution (Berger & Heath,

1968). The scale parameter of this exponential distribution is simply the mixed layer depth divided by the sedimentation rate (alternatively parameterised by rate = 1/scale). Both the mean and the standard deviation of the exponential distribution are equal to the scale, and so the ratio of the mixed depth and sedimentation rate give a prediction of the standard deviation of ages according to this model. Using a typical mixing depth of 10 cm (Boudreau, 1998) and the observed sedimentation rate of 2.2 cm/ka, we obtain a standard deviation of 10 / (2.2 / 1000) = 4545 14C years, remarkably close to the 4670 14C years from the simulation.

We noticed one other potential error. In figure 2, the calibrated ages in panel B appear to be about $\frac{1}{2}$ the 14C ages in panel A, whereas calibrated ages should be older than their 14C ages (but not by 2x).

Berger, W. H., & Heath, G. R. (1968). Vertical mixing in pelagic sediments. Journal of Marine Research, 26(2), 134–143. Boudreau, B. P. (1998). Mean mixed depth of sediments: The wherefore and the why. Limnology and Oceanography, 524–526.

---

## Author Comment (AC2) · 20 Nov 2017

Dear Andrew Dolman, Sze Ling Ho and Thomas Laepple,

Thank you for your interest in our manuscript and for your enthusiastic and encouraging words regarding our single foraminifera analysis.

We would be happy to address the concerns that you have raised in the open discussion forum. Let's start with the most straightforward one:

> *We noticed one other potential error. In figure 2, the calibrated ages in panel B appear to be about ½ the 14C ages in panel A, whereas calibrated ages should be older than their 14C ages (but not by 2x).*

Sorry for any confusion this may have caused. Something happened when scaling the figures to the correct paper size in Matlab when saving to PDF format. During this process, the tick labels (which had been manually converted from years to ka) on Fig. 2B became mislabelled when the figure was resized. The x-axis labels in Figure 2B should not read 0, 5, 10, 15, 20, 25 ka, but rather 0, 10, 20, 30, 40, 50 ka. A classic case of pre-upload figure fatigue. It's great that you informed us about this, because it would have been a shame if the figure had made it into the final version with incorrect tick labels.

Regarding your other comments:

> *Their figure 4 shows the results of the simulation. Using the observed sediment accumulation rate of 2.2 cm/ka, they estimate the standard deviation of movement required in order to obtain a SD of age of 4670 14C years as approximately 6 cm.*

We must stress here that the exact words we used were "at least 6 cm" (Line 173 in the Discussion manuscript). We considered uncertainty when representing the intercept between the sediment accumulation rate (SAR) of 2.2±0.9 cm/ka and 4670 1σ age variation of a 1 cm slice, hence the intercept is represented by a grey area in Figure 4. This grey area begins at approximately 6 cm on the x-axis, so we have written "at least 6 cm". However, it occurs to us that we could have described this grey area of uncertainty more clearly in the text and figure caption, as it may have led to a misunderstanding. We will make things clearer for the reader and find a better way to graphically represent the intercept.

> *If a constant sedimentation rate is assumed, then is the SD in depth not simply the SD in time (4670 years) scaled by the sedimentation rate, i.e. 4670 * (2.2 / 1000) = 10.274 cm?*

If we follow your calculation, but also take into account the SAR uncertainty we reported (±0.9), then one would get a range of 6.1 – 14.5 cm, consistent with our interpretation of "at least 6 cm".

> *We think this paper could be improved by instead considering a well established physical model of bioturbation in which there is a well-mixed surface layer of sediment of a fixed depth (Berger & Heath, 1968). Assuming a constant sedimentation rate, and a fixed mixing depth, the time integrated solution to this simple model predicts that the ages of material at a given depth follow an exponential distribution (Berger & Heath, 1968). The scale parameter of this exponential distribution is simply the mixed layer depth divided by the sedimentation rate (alternatively parameterised by rate = 1/scale). Both the mean and the standard deviation of the exponential distribution are equal to the scale, and so the ratio of the mixed depth and sedimentation rate give a prediction of the standard deviation of ages according to this model. Using a typical mixing depth of 10 cm (Boudreau, 1998) and the observed sedimentation rate*

*of 2.2 cm/ka, we obtain a standard deviation of 10 / (2.2 / 1000) = 4545 14C years, remarkably close to the 4670 14C years from the simulation.*

[Figure]

Our Fig. 4 in the discussion paper is consistent with the method you describe for our sediment core. Your calculation using a mixing depth model approach yielded *10 / (2.2 / 1000) = 4545 yrs* when uncertainties are not considered, which you note is similar to our calculated 1 SD single cm age value of 4670 yrs. Our own simulation shows similar behaviour for our sediment core: Above we have drawn on our Fig. 4 the intercept lines (in red) for 2.2 cm/ka on the y-axis and 10 cm on the x-axis, momentarily also not considering uncertainty. The contour region where the two lines intercept is between 4000 and 5000 yrs. So in this case there is very good agreement. However, we investigated other x-axis and y-axis values against the Berger and Heath (1968) equation, and the agreement was not quite as consistent for regions in Fig. 4 where the contour lines bend towards horizontal, so we became curious, which brings us back to another comment you made:

> *We note that in the region of the observed sediment accumulation rate for this core, the contour line for a SD of ~ 5000 14C years is more or less parallel to the PDSM axis and therefore there is little power to constrain the strength of PDSM.*

After your interesting comment above we also became curious as to why our contour lines in Fig. 4 'bend' towards horizontal. After looking through the simulation it was discovered that the bending of the lines was because the simulation was originally not provided with enough synthetic core lengths to accurately reproduce more intense redeposition of single foraminifera (i.e. further to the right on the x axis). After we reran our simulation with the necessary synthetic core lengths, we produced a new Fig. 4 (now without bending lines, see below) that is entirely consistent with the Berger and Heath (1968) equation that you describe for all regions of the figure. We will use this updated figure in the new manuscript and also mention that it fully agrees with Berger and Heath (1968).

[Figure]

We will now also mention in the conclusions that our single foraminifera post-depositional movement simulation using Gaussian noise  is consistent with existing bioturbation models and also consistent with the single foraminifera $^{14}$C age variation we have measured for our core. The use of Gaussian noise is also supported by the fact that the single foraminifera $^{14}$C results in our study suggest a normal distribution in post-depositional movement of single foraminifera. In other words, our approach reconciles both existing models of bioturbation and our single foraminifera data.

Regarding mixing depths: We didn't really want to talk about mixing depths in this manuscript, because we thought that would be overinterpretation of the data. However, we will add a couple of sentences and references for the benefit of the reader.

Hopefully we have been able to fully address your concerns. Thanks again for your helpful comments in the open discussion forum and for the interesting references for us to cite. Your input will help us to better communicate our method and results and to strengthen the manuscript.

On behalf of the co-authors,

Sincerely,

Bryan Lougheed

---

## Referee Comment (RC2) · Anonymous Referee #2 · 21 Nov 2017

The authors present an interesting approach to generating age-models that may be especially beneficial in areas of low sediment accumulation rates. They used paired C14 and stable isotope measurements on benthic foraminifera and perform a variety of sensitivity tests. I found the paper well written and have a few minor suggestions.

I think one obvious limitation is that this approach only works for samples that can be C14 dated. This should be stated.

I have no suggestions as to how to deal with the edge effect. However, omitting ∼20%

of the data to pass a K-S test for a p value <0.05 is a bit of a statistical manipulation. The CDF is a reasonable fit in Fig 3B. I'm not sure what is gained by eliminating part of the dataset to pass a statistical test with an arbitrary p-value of 0.05.

For Figure 4, it's not really clear to me how a sediment accumulation rate was calculated for LR04. LR04 is a stack of benthic foraminifera records from a variety of depositional environments. It's also an interpolated stack so the sampling resolution is variable.

Line 56 = change utilise to use

Line 71 = Please explain how the planktic record showed significant PDSM. It didn't have an acceptable stable-isotope stratigraphy?

Line 122 = "Age uncertainties concealed by the current state-of-the-art" is an odd phrasing. I would use Age uncertainties obscured by conventional geochronological techniques or something like that. I would say the method described in the paper is more "state-of-the-art" than the normal techniques.

Line 125 = Following along the lines as the comment above, "we show that such significant PDSM can be concealed by the current geochronological techniques.." State-of-the-art implies slightly something different that you're more or less advocating with your own technique.

Line 160 = "edge effect"

---

## Author Comment (AC3) · 23 Nov 2017

Dear Julia Gottschalk,

Thank you for interest in our manuscript, your encouraging words and for your contribution to the discussion forum of this manuscript.

Indeed, when analysing for the rare radioisotope 14C on such small samples, one must be vigilant for any possibility of modern 14C contamination, as pointed out in the publications you have cited in your short comment: Brown and Southon (1997); Hua

et al. (2004). Those publications specifically refer to possible contamination during the graphitisation process, which we did not use. The method which we use involves acidifying sample carbonate to sample $CO_2$ in He-flushed, sealed septa vials for direct measurement in the MICADAS AMS (e.g. Wacker et al., 2013). This method has much smaller sample mass requirements than the graphitisation process, meaning that it is possible to measure much smaller sample masses on the AMS. However, we agree that the general principle of small sample masses being relatively more susceptible to contamination of course applies, just that the mass threshold is much lower than in the case of graphitisation. We measured IAEA-C1 standard powder, as well as single Eemian (i.e. assumed 14C "blank") foraminifera from the same species and sediment core. The masses of these blank foraminifera samples (min: 12.8 $\mu$g C, max: 24.0 $\mu$g C, mean: 19.0 $\mu$g C) were of similar range to that of the foraminifera used for the study (min: 8.4 $\mu$g C, max: 47.2 $\mu$g C, mean: 19.8 $\mu$g C). Procedural blank foraminifera values ranged between ∼31,000 and ∼41,000 14C yr BP and were generally not as good as those for IAEA-C1 material. There was no correlation between sample mass and 14C value, neither in the case of the IAEA-C1 material nor the blank forams.

The specific blank correction value used was F14C = 0.0095 $\pm$ 0.00203 (∼37500 14C yr BP). This blank value, along with its uncertainty, was applied using the BATS software (Wacker et al., 2010). For a smaller, second run of samples (ETH-74XX) it was possible to use a much lower blank (F14C = 0.0033 $\pm$ 0.00101).

You are correct that the variability of process blanks and the reduced measurement sensitivity reduce, for now, for now, the ability to precisely measure older samples. Moreover, as you suggest, the variation also reduces the precision somewhat on younger dates (e.g., in the <20,000 14C yr BP range). Compared to a constant blank value of 37,500 14C yr, variable blank values between ∼31,000 and ∼41,000 14C yr would affect 14C ages of Holocene samples (i.e. the bulk of our samples) by between ∼0.9% and ∼3.5%, thus increasing uncertainty. Due to the relatively poorer counting statistics inherent in 14C dating single forams, our reported 14C dates already have

quite a low precision (between 1% and 4%) anyway. Moreover, the uncertainties associated with 14C reservoir age reduce chronological precision by a further 7.5% on average. Therefore, the variation in blank value is not significant in determining the quantified PDSM in T86-10P (which is on the order of thousands of years). In general, dating single forams sacrifices precision to gain accuracy. Such an approach is especially useful in sediment cores with high PDSM and/or very low SAR, where large multi-specimen samples would yield a good 14C signal with low error, but all information regarding the large intra-sample 14C age heterogeneity would be lost.

Pre-treatment options for single forams is something that we are continually looking into. It is our intention that in future, a routine and (automated) prep system for single foraminifera will be developed, with the ability to carry out preatreatment online and split gaseously into stable isotope and 14C fractions. However, this depends on access to project funding.

Thank you also for pointing out that no information is provided regarding sample size (i.e. mass) in the supplemental table. We agree that this information should definitely be provided in the case of our study, as we detail new methods. We will include information regarding sample ug C as reported by the MICADAS AMS in the supplemental table. You might note that some of the larger sample masses with 14C data have no corresponding stable isotope data. In principle it should have been possible to also report stable isotope data for these larger samples, but we suffered a failed IRMS run at another laboratory (not at our affiliation institutions).

Once again, thank you for your interest in our manuscript and specifically your helpful comments requesting more information about the blank correction process. We will better highlight this process (as explained above) in the updated version of the manuscript, which will certainly serve to significantly improve the reader experience.

On behalf of the co-authors, kind regards,

Bryan Lougheed

Wacker, L., Christl, M. and Synal, H.A., 2010. BATS: a new tool for AMS data reduction. Nuclear Instruments and Methods in Physics Research Section B: Beam Interactions with Materials and Atoms, 268(7), pp.976-979.

Wacker, L., Lippold, J., Molnár, M. and Schulz, H., 2013. Towards radiocarbon dating of single foraminifera with a gas ion source. Nuclear Instruments and Methods in Physics Research Section B: Beam Interactions with Materials and Atoms, 294, pp.307-310.

---

## Author Comment (AC4) · 23 Nov 2017

Dear Reviewer,

Thank you for providing a review for our manuscript and for your positive words concerning the potential of the method we have developed. We would be happy to briefly discuss your comments below:

"I think one obvious limitation is that this approach only works for samples that can be C14 dated. This should be stated."

It is stated in the title of the manuscript that the method involves radiocarbon. However, we will further underline in the manuscript that the method is best suited for late-glacial and Holocene samples (i.e. those within the suitable age range for the radiocarbon method.

"I have no suggestions as to how to deal with the edge effect. However, omitting âĹij20% of the data to pass a K-S test for a p value <0.05 is a bit of a statistical manipulation. The CDF is a reasonable fit in Fig 3B. I'm not sure what is gained by eliminating part of the dataset to pass a statistical test with an arbitrary p-value of 0.05."

As you point out, in Fig. 3B the CDF appears normal upon first viewing. We omit the data from the 20 youngest and 20 oldest foraminifera in Fig 3C to demonstrate the presence of the temporal 'edge effect' upon our data. It would likely be possible to omit much less of the 'edge data' and still pass the K-S normality test: the amount omitted was not chosen with the express intent of passing the K-S test wit p=0.05. In any case, we show both Fig. 3B and Fig. 3C for the benefit of the reader.

"For Figure 4, it's not really clear to me how a sediment accumulation rate was calculated for LR04. LR04 is a stack of benthic foraminifera records from a variety of depositional environments. It's also an interpolated stack so the sampling resolution is variable."

We had calculated a SAR of 3.8 $\pm$ 0.9 cm/ka for LR04, based on the mean and standard deviation of the LR04 average sedimentation rate provided by Lisiecki and Raymo (2005). We will try to include error bars on the figure, or at least mention in the text how the calculation was carried out.

"Please explain how the planktic record showed significant PDSM. It didn't have an acceptable stable-isotope stratigraphy?"

That is correct. We will provide more information about this.

Thank you once again for your comments and your review, which will help to improve the manuscript.

On behalf of the co-authors,

Kind regards,

Bryan Lougheed
* * *

---

## Author Comment (AC5) · 29 Nov 2017

Dear Philippa Ascough,

Thank you for your extensive review of our manuscript and for the many constructive suggestions. Apologies for the delayed reply. This manuscript generated a number of comment contributions in the discussion forum. Time had to be found to address them all.

We would be happy to address your main discussion points below:

[Figure]

"In a composite date, if the preponderance of measurements were representative of the age of the sediment (albeit with a long tail), this would increase the accuracy of the date, as the inaccurate ages would contribute proportionally less to the composite age. Can the authors comment on any statistical manipulation that could be used in this instance to improve accuracy? Also, how would one properly calculate (even semi-quantitatively) the uncertainties associated with composite measurements? Can the authors offer any suggestions for this? It would be good to see these points considered within the manuscript."

Our main suggestion for improving the accuracy of a multi-specimen date (e.g. existing conventional methods) would be to simply propagate extra uncertainty into the 14C determination in order to take into account the large intra-sample age heterogeneity. This could be done using our Figure 4 as a guide, or by considering bioturbation models suggested by you below or by the other contributors to the discussion panel. We will highlight this better in the revised manuscript.

"The addition of older material would have a proportionally smaller effect on the measured age than the addition of (much) younger material. Can the authors comment on how this would affect how the precision on measurements should be calculated (mass balance approach?)"

Very interesting comment. Indeed, it is additionally possible that multi-specimen foraminifera samples can be biased towards younger ages. We agree that researchers should additionally take this effect into account when considering uncertainty for multi-specimen foraminifera dates. We will mention this in the manuscript.

"One interesting point is that perhaps the changes in PDSM that could be identified with this approach could tell us something about sediment dynamics. I appreciate this is a little outside the scope of the paper, but could be mentioned as a 'silver lining'. One reference that would be good to include is: Berger, W.H. and Johnson, R.F. 1978: On the thickness and the 14C age of the mixed layer in deep sea carbonates. Earth

and Planetary Science Letters 41, 223–27. The fiĄndings in this support the authors results"

We didn't want to go too much into detail regarding sediment dynamics in this manuscript, but it does seem (as discussion contributors Dolman et al also subsequently suggested) that our data is in good agreement with the aforementioned bioturbation studies. We agree that it would indeed be a 'silver lining' if we were to go one step ahead and mention this in the manuscript. We will do so.

"How many samples would the authors advocate measuring in order to get a decent idea of the true amount of sample heterogeneity?"

The most straightforward method would be to date as many foraminifera as possible from the same 1 cm interval and analyse the age distribution from that interval. However, the problem with such an approach is that there is only a limited number of foraminifera of sufficient mass within any given sediment interval. Instead, we analysed as many as possible from multiple intervals and analysed PDSM by reconstructing the post-depositional ranking change of the single foraminifera (i.e. age ranking vs depth ranking). It is difficult to quantify the minimum number of necessary foraminifera to carry out a successful PDSM analysis using this method because the resulting age distribution of the foraminifera picked from the various levels is to a certain extent dependent upon luck (i.e. when PDSM is severe or SAR is low, one does not know the approximate age of the foraminifera that are being picked).

"Were the forams pretreated in any way? i.e. washing/ agitation with distilled water, or removal for surface C by preliminary acid dissolution?"

Acid pretreatment was only possible for select, larger foraminifera. In these cases, the 14C age of both the initial acid leach and remaining foraminifera was investigated. Foraminifera were washed during the wet sieving and subsampling process, but not again just prior to measurement. We are currently investigating various methods for pre-treatment on such small samples, project funding permitting.

You also asked about the blank correction. For extensive information regarding the blank correction process, we refer to our reply to Julia Gottschalk.

Thanks again for your helpful review. We look forward to using your input to improve the final version of our manuscript.

On behalf of the co-authors,

Sincerely,

Bryan Lougheed

---

## Author Response (AR1)

Dear Alessio Rovere,

Thank you again for reviewing our manuscript and for allowing for an extension to the revision deadline. I was a bit busy due to having moved countries since the manuscript was originally submitted and the extension was much welcomed!

The open discussion format of *Climate of the Past* has allowed us to process some very interesting comments by the reviewers and discussion contributors. The main changes to the manuscript relate to the consideration of bioturbation upon the core material. Following the helpful comments of reviewer Ascough and discussion contributors Dolman et al., we decided to simulate multiple bioturbation scenarios and how they relate to the single foraminifera $^{14}$C data that we have found. In the previous manuscript, we proposed that normally distributed age-depth sediment relationship could explain the bioturbation we see, whereas Dolman et al pointed out that the traditional bioturbation model of Berger and Heath (1968) would predict an exponential age-depth depth relationship. After carrying out multiple sediment core simulations, and also considering other forms of bioturbation, we have found that the observed PDSM in core T86-10P can best be explained by multiple bioturbation processes, which neither a normally distributed nor exponentially distributed age-depth relationship can fully explain. Specifically, we find that we can reproduce the bioturbation in T86-10P when we simulate much deeper, secondary bioturbation processes documented by (Löwemark and Grootes, 2004; Löwemark and Werner, 2001; Löwemark and Schäfer, 2003). These secondary processes are not included in traditional bioturbation models (e.g., those by Berger and Heath 1968, Berger and Johnson, 1978; Berger and Killingley, 1982; Guinasso and Schink, 1975; Peng et al., 1979). We have, therefore, created our own bioturbation model which combines the primary bioturbation processes predicted by traditional bioturbation models with the secondary, much deeper *Zoophycos* bioturbation. We find that the combination of these two types of bioturbation can reconcile the single foraminifera $^{14}$C data that we see in core T86-10P. We also felt it would be better to replace the contour plot simulation output in the original submission with a more reader friendly figure which visually shows the age-depth distribution for our single foraminifera simulations.

The changes we outline are slightly more extensive than those we originally outlined in the response to the reviewers, but we feel they are worthwhile as they reconcile the observed single foraminifera $^{14}$C data in core T86-10P by integrating multiple models of bioturbation. We note that our main end product of the manuscript, the dual $^{14}$C and stable isotope analysis, remains unchanged from the original manuscript.

We feel that the *Climate of the Past* open discussion forum has served to improve our manuscript and affirms our choice in submitting to this journal. We'd like to once again warmly thank the reviewers and discussion contributors.

We thank you for considering our manuscript and hope you find the improved manuscript to be suitable for publication in *Climate of the Past*. Appended to this letter is the 'track-changes' version of the manuscript (which MS Word has somewhat exaggerated!).

On behalf of the co-authors,

Kind regards,

Bryan Lougheed

[revised manuscript text omitted]

---

## Referee Report (RR1)

Dear Editors,

I read and commented on the initial version of the manuscript, which was already a very interesting and useful paper. The main point in our comment was that, to a first approximation, the observed heterogeneity in $^{14}C$ ages could be predicted from the sediment accumulation rate, typical bioturbation depths, and an established simple physical model of bioturbation. One of the potential applications of their method is as an independent test of these simple bioturbation models.

In this revised version the authors now consider their results in the context of this simple bioturbation model and also consider an expanded model that would features of the data not explained by the simple model. In addition to this, several other minor issues have been dealt with in this revised version.

In my opinion the paper is ready to be published and would be a perfect fit for Climate of the Past.

Dr Andrew M. Dolman.